

# Bidirectional correlation between gastroesophageal reflux disease and sleep problems: a systematic review and meta-analysis

Xiaolong Tan[1], Shasha Wang[2], Fengjie Wu[1] and Jun Zhu[1]

[1] Department of Gastrointestinal Surgery, Binzhou Medical University Hospital, Binzhou, Shandong Province, China
[2] Department of Oncology, The People's Hospital of Binzhou City, Binzhou, Shandong Province, China

## ABSTRACT

**Objectives:** Gastroesophageal reflux disease (GERD) and sleep problems are highly prevalent among the general population. Both them are associated with a variety of psychiatric disorders such as depression and anxiety, which is highlighting an underexplored connection between them. This meta-analysis aims to explore the association between sleep problems and GERD.

**Methods:** We conducted a comprehensive search on PubMed, Cochrane Library, Embase, and Web of Science, using Medical Subject Headings (MeSH) and keywords, covering articles from the inception of the databases until August 2023. Stata statistical software, version 14.0, was utilized for all statistical analyses. A fixed-effects model was applied when $p > 0.1$ and I2 ≤ 50%, while a random-effects model was employed for high heterogeneity ($p < 0.1$ and I2 > 50%). Funnel plots and Egger's test were used to assess publication bias.

**Results:** Involving 22 studies, our meta-analysis revealed that insomnia, sleep disturbance, or short sleep duration significantly increased the risk of GERD (OR = 2.02, 95% CI [1.64–2.49], $p < 0.001$; $I^2$ = 66.4%; OR = 1.98, 95% CI [1.58–2.50], $p < 0.001$, $I^2$ = 50.1%; OR = 2.66, 95% CI [2.02–3.15], $p < 0.001$; $I^2$ = 62.5%, respectively). GERD was associated with an elevated risk of poor sleep quality (OR = 1.47, 95% CI [1.47–1.79], $p < 0.001$, $I^2$ = 72.4%), sleep disturbance (OR = 1.47, 95% CI [1.24–1.74], $p < 0.001$, $I^2$ = 71.6%), or short sleep duration (OR = 1.17, 95% CI [1.12–1.21], $p < 0.001$, $I^2$ = 0).

**Conclusion:** This meta-analysis establishes a bidirectional relationship between four distinct types of sleep problems and GERD. The findings offer insights for the development of innovative approaches in the treatment of both GERD and sleep problems.

Corresponding author
Jun Zhu, byfyzhujun@163.com

## INTRODUCTION

As the pace of life quickens, sleep problems are becoming increasingly prevalent. with an article reporting a prevalence rate of 7% (*Liu et al., 2016*). The digestive system is particularly sensitive to lifestyle changes due to its connection to emotions, resulting in an increase in gastrointestinal disorders such as gastroesophageal reflux disease. Disrupted circadian rhythms due to sleep problems can impact melatonin secretion, potentially leading to depression and anxiety (*De Berardis et al., 2013*), factors that may exacerbate GERD incidence (*Zamani et al., 2023*). Moreover, a link has been found between GERD and sleep problems; individuals with either GERD (*Hu et al., 2024*) or sleep problems (*Shoib et al., 2022*) are more likely to experience obstructive sleep apnea.

Gastroesophageal reflux disease (GERD), is a condition triggered by the reflux of gastric contents into the esophagus, with its diagnosis being based on typical symptoms or visible mucosal damage observed during endoscopy (*Sasankan & Thota, 2022*). As its prevalence rises, further investigation becomes imperative (*Peery et al., 2019*). Despite recent advancements in our understanding of its pathology, drug development, and treatment methods, optimal patient outcomes are yet to be achieved. This underscores the urgency to augment clinician awareness of GERD-related symptoms for early diagnosis, and addressing early risk factors may be key to preventing the development of GERD. Prior meta-analyses confirm higher GERD prevalence in obese, smoking, and NSAID individuals (*Eusebi et al., 2018*). Additionally, a correlation has been identified linking reflux to apnea, reduced sleep efficiency, and decreased oxygen levels during sleep (*El Hage Chehade et al., 2023*), which warrants further exploration of sleep issues and their connection to GERD.

Sleep problems, ranging from insomnia, short duration, disturbances, and poor quality, are encountered among the general population and are associated with a broad array of health complications, such as lung disease (*Sunwoo & Owens, 2022*), high blood pressure (*Ziegler, 2003*), cardiovascular conditions (*Pomeroy et al., 2023*), migraines (*Bigal & Lipton, 2006*), cognitive decline (*Sun et al., 2023*), and mental disorders (*On et al., 2017*). Insufficient sleep can precipitate abnormal acid exposure in the esophagus (*Yamasaki, Quan & Fass, 2019*), a significant risk factor for GERD due to prolonged acid exposure (*Hung et al., 2016*). GERD is particularly linked to insomnia (*Suganuma et al., 2001*), and this relationship forms the basis of our hypothesis: we propose a potential bidirectional relationship between GERD and sleep problems, and to examine this, we conducted a methodical review of population-based evidence to elucidate their association.

## METHOD

This study was performed in accordance with the Preferred Reporting Items for Systematic Evaluation and Meta-Analysis 2020 (PRISMA, 2020) guidelines (*Page et al., 2021*). The protocols have been pre-registered with the International Prospective Register of Systematic Reviews (PROSPERO) platform under the approval number: CRD42023452348.
## Data sources

We retrieved publicly accessible studies up to August 2023 from PubMed, Cochrane Library, Embase and Web of Science. The language is restricted to English. The search strategy was a combination of medical subject headings (Mesh) and text words. The keywords used for the search were 'gastro-esophageal reflux', 'gastric acid reflux', 'gastric acid reflux disease', 'gastro-esophageal reflux disease', 'reflux disease, gastro-esophageal' as well as 'sleep*'. All search terms used for the retrieval of articles are detailed in Tables S3–S6.

## Eligibility criteria

We included case-control or cohort studies that assessed the association between gastroesophageal reflux disease (GERD) and sleep problems. According to the Montreal definition, GERD is a condition which develops when the reflux of stomach contents causes troublesome symptoms and/or complications (*Vakil et al., 2006*). A diagnosis of GERD could be made clinically by any of the following: (A) heart burn and/or regurgitation of any severity, or symptoms felt to be compatible with gastroesophageal reflux as diagnosed by a clinician or according to a questionnaire; (B) esophageal erosions defined by endoscopy. Sleep problems almost always can be diagnosed based solely on a careful history. Therefore, after reviewing the literature, we have identified four categories of sleep problems, including: sleep disturbance, short sleep duration, insomnia and poor sleep quality. In this study, sleep disturbance means people were found to be struggling to fall asleep, or waking up too early and not being able to get back to sleep. Criteria for insomnia include difficulty initiating or maintaining sleep, according to the Diagnostic and Statistical Manual of Mental Disorders, 4th edition (DSM-IV) or self-report. Short sleep duration was defined as sleeping less than 7 h on average per night. The Pittsburgh Sleep Quality Index or a Likert scale containing the question: how do you rate your sleep quality is used to assess sleep quality. Poor sleep quality is considered to be present if the patient's Pittsburgh Sleep Quality Index is higher than normal or if the patient reports that sleep quality is poor.

The articles included had to be fulfilled the following criteria: (1) case-control or cohort study; (2) investigations of the association of gastroesophageal reflux with the risk of incident any type of sleep problem *vice versa*; (3) provide an odds ratio (OR) estimate with corresponding 95% confidence interval (CI). The exclusion criteria included the following: (1) Studies did not provide an odds ratio (OR) estimate with 95% confidence interval (CI). (2) Literature with the same data. (3) Conference abstracts, study protocols, duplicate publications and studies without outcomes of interest.

## Study selection

Study selection was performed by two reviewers (XLT and SSW) who independently screened the literature based on the eligibility and exclusion criteria. Duplicate and irrelevant articles were first excluded from the titles and abstracts. The full text of potentially eligible articles was then downloaded and read to identify all eligible studies. Any disagreements were resolved by the third reviewer (WFJ), who acted as an arbiter.

## Data extraction

Two reviewers (XLT, FJW) independently extracted the following information according to the guideline for data extraction for systematic reviews and meta-analysis (*Taylor, Mahtani & Aronson, 2021*), including the following information: first author, study type, country, year of publication, age of participants, sample size, diagnosis of GERD and different type of sleep problem, type of sleep problem, confounder, odds ratio and 95% confidence interval.

## Risk of bias

To ensure a comprehensive assessment, risk of bias was evaluated using the Newcastle-Ottawa Scale (NOS) (*Glaviano, Bazett-Jones & Boling, 2022*) by classifying studies as either case-control or cohort studies. The NOS tool awards stars to responses meeting the eligibility criteria, a maximum total of nine stars can be attained by each study: four for selection, two for comparability, and three for outcome, with a higher star count reflective of a superior study quality. Scores of 0–3, 4–6, and 7–9 were regarded as indicative of low, moderate, and high quality, respectively.

## Statistical analysis

The adjusted OR and 95% CI from each study were used to assess the association between GERD and sleep problems. The $\chi^2$ test and $I^2$ values were used for the assessment of heterogeneity. A fixed effects model was used when $p > 0.1$ and $I^2 \leq 50\%$. If $p < 0.1$ and $I^2 > 50\%$ indicated high heterogeneity (*Lei et al., 2022*), a random effects model was used (*Higgins et al., 2003*). To check the robustness of the overall effects, the sensitivity analysis was performed by excluding one study each time and rerunning the analysis. Publication bias was confirmed by visual inspection of funnel plots and statistical assessment using Egger's regression test (*Egger et al., 1997*). We performed several analyses based on GERD and each type of sleep problem. All statistical analyses were carried out using the Stata statistical software package, version 14.0 (StataCorp, College Station, TX, USA).

# RESULTS

## Literature search

During the literature search, a total of 5,962 records were identified from the above databases. The first step was to exclude 973 duplicate articles. In the second step, after screening the abstracts and titles, 4,589 records were excluded. Subsequently, we excluded meta-analyses and systematic reviews. Finally, 22 studies (*Ahmed et al., 2020*; *Cadiot et al., 2011*; *Chang et al., 2021*; *Chen et al., 2009*; *Cremonini et al., 2009*; *Emilsson et al., 2022*; *Fass et al., 2005*; *Ha et al., 2023*; *Horsley-Silva et al., 2019*; *Hyun, Baek & Lee, 2019*; *Jansson et al., 2009*; *Ju et al., 2013*; *Lei et al., 2019*; *Lindam et al., 2012, 2016*; *Murase et al., 2014*; *Okuyama et al., 2017*; *Wallander et al., 2007*; *Yadegarfar et al., 2018*; *You et al., 2015*; *Zhang et al., 2012a, 2012b*) were included in our meta-analysis after excluding literature with unrelated outcomes, conference abstracts, and literature from which data could not be extracted. The selection process is illustrated in the Fig. 1.
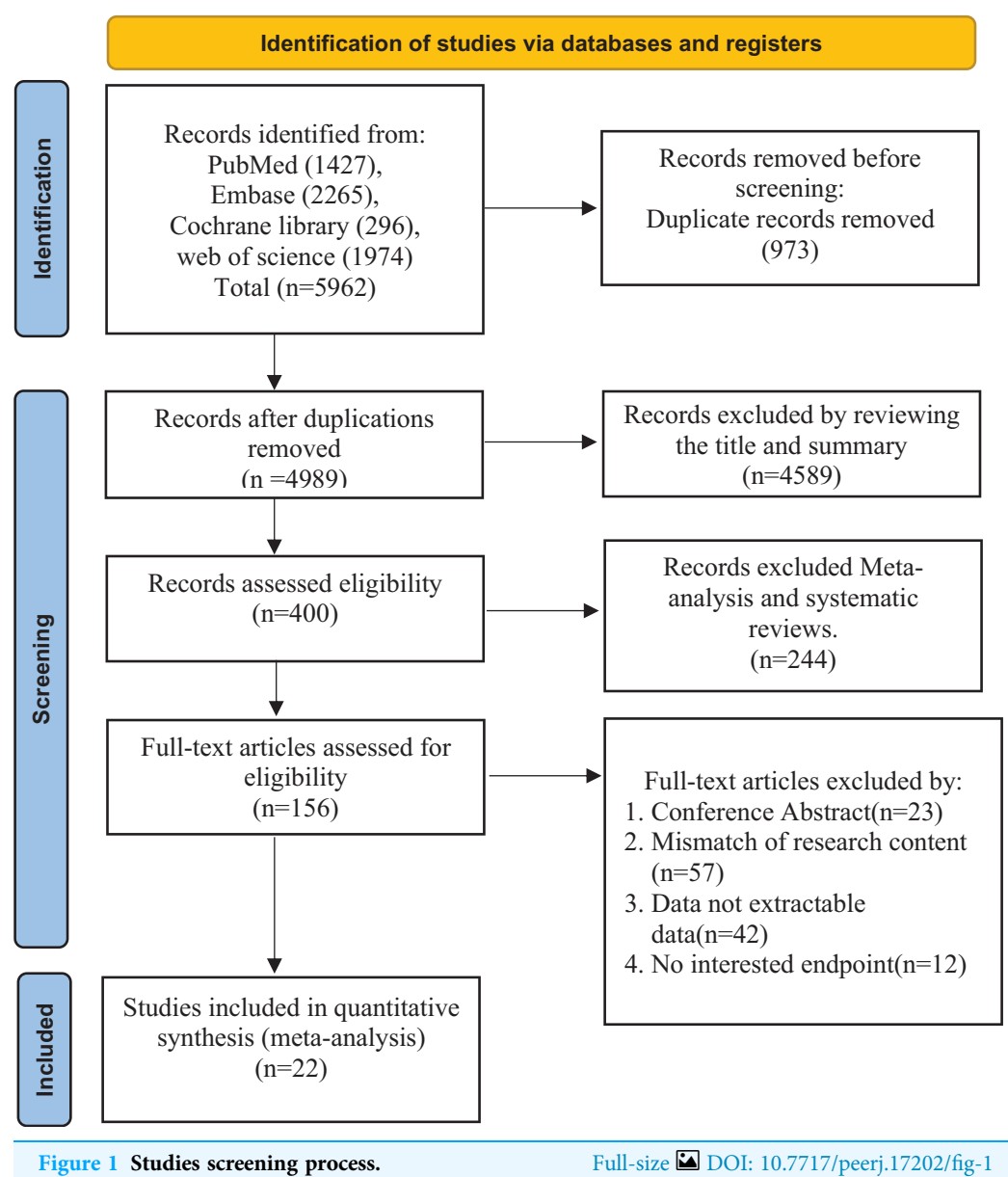

**Figure 1  Studies screening process.**               

## Study characteristics

Of the 22 studies (*Ahmed et al., 2020*; *Cadiot et al., 2011*; *Chang et al., 2021*; *Chen et al., 2009*; *Cremonini et al., 2009*; *Emilsson et al., 2022*; *Fass et al., 2005*; *Ha et al., 2023*; *Horsley-Silva et al., 2019*; *Hyun, Baek & Lee, 2019*; *Jansson et al., 2009*; *Ju et al., 2013*; *Lei et al., 2019*; *Lindam et al., 2012, 2016*; *Murase et al., 2014*; *Okuyama et al., 2017*; *Wallander et al., 2007*; *Yadegarfar et al., 2018*; *You et al., 2015*; *Zhang et al., 2012a, 2012b*) that were included, 14 were case-control and eight were cohort studies, spanning from 2005 to 2023. Table 1 (*Ahmed et al., 2020*; *Chang et al., 2021*; *Emilsson et al., 2022*; *Fass et al., 2005*; *Jansson et al., 2009*; *Ju et al., 2013*; *Lindam et al., 2012, 2016*; *Yadegarfar et al., 2018*; *Zhang et al., 2012a, 2012b*) presents the characteristics of the 11 included studies with GERD as an outcome. In these studies, risk factors identified for GERD included sleep disturbance, short sleep

**Table 1 Basic information on the included literature with gastroesophageal reflux disease as an outcome.**

| Author | Year | Country | Study type | Exposure size | Normal size | Age (years) | Sleep problem type | Confounders adjusted |
|---|---|---|---|---|---|---|---|---|
| *Jansson et al.* | 2009 | Norway | Case-control | 3,153 | 40,210 | 19–81+ | Insomnia, Sleep disturbance | Age, sex, smoking, BMI, SES, anxiety, depression, myocardial infarction, angina pectoris, stroke, nausea, diarrhea, constipation |
| *Emilsson et al.* | 2022 | Sweden | Cohort study | 839 | 4,872 | Exposure: 29–57; Normal: 39–67 | Short Sleep Duration | Age, BMI, smoking status, caffeine consumption, alcohol dependence, physical activity level, depression, anxiety, snoring |
| *Lindam et al.* | 2012 | Sweden | Case-control | 1,327 | 6,687 | 65–75+ | Insomnia, Sleep disturbance | Age, sex, educational level, BMI, smoking |
| *Zhang et al.* | 2012a | Hong Kong | Cohort study | 185 | 2,106 | Mean (SD): 41.1 (5.4) | Sleep disturbance | Age, gender, education level, marital status, family income, regular use of medication(s), subtypes of insomnia, snoring, sleep duration |
| *Ahmed et al.* | 2020 | Pakistan | Case-control | 1,000 | 1,000 | Exposure: mean (SD): 30 (10.47); Normal: mean (SD): 44.73 (13.92) | Short Sleep Duration | Age, gender, BMI |
| *Lindam et al.* | 2016 | Norway | Cohort study | Total: 16,754 | | Mean (SD): 43 (12) | Insomnia, Sleep disturbance | Sex, age, BMI, smoking, education, anxiety, depression |
| *Yadegarfar et al.* | 2018 | Iran | Case-control | 717 | 308 | Exposure: mean (SD): 39.1 (9.6) Normal: mean (SD): 39.93 (10.7) | Short Sleep Duration | |
| *Zhang et al.* | 2012b | Hong Kong | Cohort study | 115 | 2,036 | Mean (SD): 46.3 (5.1) | Insomnia | Age, gender, education level, family income, regular use of medication |
| *Fass et al.* | 2005 | USA | Cohort study | 6,369 | 15,699 | Exposure: mean (SD): 62.9 (10.9) Normal: mean (SD): 63.6 (10.4) | Insomnia | |
| *Ju et al.* | 2013 | Korean | Case-control | 21 | 513 | Exposure: mean (SD): 50.8 (13.69) Normal: mean (SD): 50.95 (13.51) | Insomnia | Age, sex, alcohol consumption, BMI, depressed mood |
| *Chang et al.* | 2021 | Taiwan | Case-control | 401 | 2,249 | ≤30 3.4% 31–60 71.6% >60 25.0% | Sleep disturbance | |

duration and insomnia. Eight of these articles (*Ahmed et al., 2020*; *Emilsson et al., 2022*; *Jansson et al., 2009*; *Ju et al., 2013*; *Lindam et al., 2012, 2016*; *Zhang et al., 2012a, 2012b*) adjusted for confounders such as sex and age. Table 2 (*Cadiot et al., 2011*; *Chen et al., 2009*; *Cremonini et al., 2009*; *Ha et al., 2023*; *Horsley-Silva et al., 2019*; *Hyun, Baek & Lee, 2019*; *Lei et al., 2019*; *Murase et al., 2014*; *Okuyama et al., 2017*; *Wallander et al., 2007*; *You et al., 2015*), outlines basic information from the literature covering sleep problems, which

**Table 2 Basic information on the included literatures with three types of sleep problem as an outcome.**

| Author | Year | Country | Study type | Exposure size | Normal size | Age (years) | Sleep problem type | Confounders adjusted |
|---|---|---|---|---|---|---|---|---|
| *Horsley-Silva et al.* | 2019 | USA | Case-control | Total 16,754 | | Mean (SD): 59 (14) | Poor Sleep Quality | Age, sex, BMI, narcotic, antidepressant use |
| *Ha et al.* | 2023 | USA | Cohort study | 7,726 | 28,911 | 48–69 | Poor Sleep Quality; Sleep disturbance; Short sleep duration | Age, BMI, menopausal status or menopausal hormone use, smoking status, race, presence of cancer, congestive heart failure, diabetes, asthma, hyperthyroidism, hypothyroidism, depression, self-reported depression and anxiety symptoms, urinary incontinence, hot flushing, alcohol consumption, intake of caffeinated beverage, decaffeinated beverage, physical activity, diuretics use, proton pump inhibitor histamine-2 receptor antagonist use |
| *Okuyama et al.* | 2017 | Japan | Case-control | 483 | 1,253 | Exposure: mean (SD): 59.8 (12.1) Normal: mean (SD): 61.6 (2.0) | Sleep disturbance | |
| *Hyun, Baek & Lee* | 2019 | Korea | Case-control | 844 | 4,948 | Exposure: mean (SD): 61.99 (9.64) Normal: mean (SD): 64.06 (10.15) | Sleep disturbance | Gender, age, marital status, education level, tobacco, alcohol, physical activity, obesity, abdominal pains, heartburn, acid regurgitation, sucking sensations in the epigastrium, nausea and vomiting, borborygmus, abdominal distension and eructation |
| *Chen et al.* | 2009 | Taiwan | Case-control | 653 | 3,010 | Mean (SD): 50.6 (11.83) | Poor sleep quality; Short sleep duration | |
| *Murase et al.* | 2014 | Japan | Cohort study | Total: 8,614 | | Mean (SD): 56 (13) | Short sleep duration | |
| *Wallander et al.* | 2007 | U. K | Case-control | 12,437 | 18,350 | 20–79 | Sleep disturbance | Gender, age, Smoking status, BMI, alcohol consumption |
| *You et al.* | 2015 | Taiwan | Cohort study | 3,813 | 15,252 | 35–65.7 | Sleep disturbance | Age, sex, hypertension, diabetes mellitus, dyslipidemia, coronary artery disease, congestive heart failure, cerebrovascular disease, chronic pulmonary disease, malignancy, income and urbanist |
| *Lei et al.* | 2019 | Taiwan | Case-control | 956 | 1,718 | Exposure: mean (SD): 53.33 (11.3) Normal: mean (SD): 52.04 (10.98) | Sleep disturbance | |
| *Cremonini et al.* | 2009 | U.S.A | Case-control | 542 | 2,686 | Exposure: mean (SD): 51.5 (0.7) Normal: mean (SD): 53 (0.3) | Sleep disturbance | Age, gender, smoking status, alcohol use, mental health status score |
| *Cadiot et al.* | 2011 | France | Case-control | Total 33,391 | | | Sleep disturbance | |

includes poor sleep quality, short sleep duration, and sleep disturbance. Of the studies (*Cadiot et al., 2011*; *Chen et al., 2009*; *Cremonini et al., 2009*; *Ha et al., 2023*; *Horsley-Silva et al., 2019*; *Hyun, Baek & Lee, 2019*; *Lei et al., 2019*; *Murase et al., 2014*; *Okuyama et al., 2017*; *Wallander et al., 2007*; *You et al., 2015*) focusing on sleep problems as an outcome, six (*Cremonini et al., 2009*; *Ha et al., 2023*; *Horsley-Silva et al., 2019*; *Hyun, Baek & Lee, 2019*; *Wallander et al., 2007*; *You et al., 2015*) controlled for confounders such as gender, age, and drinking history.

## Quality assessment

According to NOS criteria, the quality score of cohort studies ranged from five to nine, with an average score of 6.55 (Table S1). Out of the included articles, 16 scored between six and eight, and only one article achieved the maximum score of 9. This suggests that the bulk of the studies included in the meta-analysis were deemed moderate to high quality.

## Insomnia and risk of GERD

We investigated the relationship between insomnia and GERD risk in six trials (*Fass et al., 2005*; *Jansson et al., 2009*; *Ju et al., 2013*; *Lindam et al., 2012, 2016*; *Zhang et al., 2012b*), which included three cohort studies and three case-control studies. The aggregated data revealed that a history of insomnia was associated with an increased risk of GERD in the pooled analysis (OR = 2.02, 95% CI [1.64–2.49], $p < 0.001$; $I^2 = 66.4\%$, z = 6.62; Fig. 2.1). Sensitivity analysis showed that none of the individual studies had a reversal of the pool effect size. That means the results are robust (Fig. S1).

## Sleep disturbance and risk of GERD

The association between sleep disturbance and GERD, which was analyzed in five studies (*Chang et al., 2021*; *Jansson et al., 2009*; *Lindam et al., 2012, 2016*; *Zhang et al., 2012a*), was highly significant. The OR was 1.98 (95% CI [1.58–2.50], $p < 0.001$) in these trials that looked at the relationship between sleep disturbance and GERD (Fig. 2.2). $I^2$ of the meta-analysis was 50.1% (z = 5.84). Sensitivity analysis upheld the reliability of these findings, showing no individual study caused a significant change in the pooled effect size, which validates that the results are robust (Fig. S2).

## Short sleep duration and risk of GERD

Out of the studies selected for evaluating the association between GERD and short sleep duration, three (*Ahmed et al., 2020*; *Emilsson et al., 2022*; *Yadegarfar et al., 2018*) showed significant results (OR = 2.66, 95% CI [2.02–3.15], $p < 0.001$; $I^2 = 62.5\%$, z = 6.92; Fig. 2.3). Sensitivity analysis confirmed the robustness of the results, as no individual study caused a reversal of the pooled effect size (Fig. S3).

## GERD and risk of poor sleep quality

During the meta-analysis concerning GERD and poor sleep quality, three publications (*Chen et al., 2009*; *Ha et al., 2023*; *Horsley-Silva et al., 2019*) yielded significant findings (OR = 1.47, 95% CI [1.47–1.79], $p < 0.001$; $I^2 = 72.4\%$, z = 3.89; Fig. 3.1). Sensitivity analysis

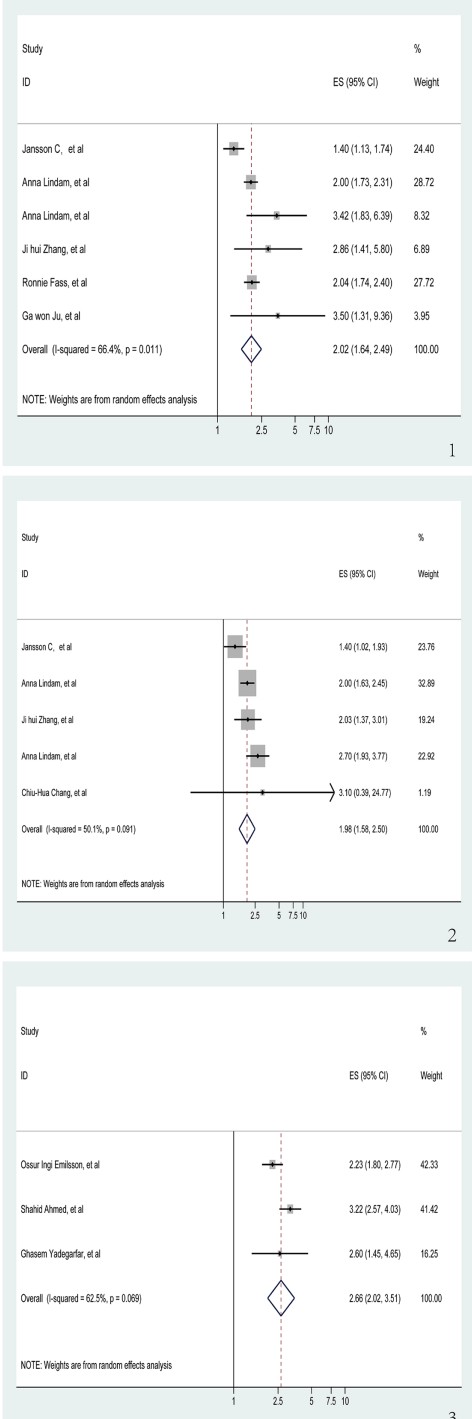

**Figure 2 Meta-analysis of the risk of GERD associated with sleep problems.** (1) Meta-analysis of the risk of GERD associated with insomnia. (2) Meta-analysis of the risk of GERD associated with sleep disturbance. (3) Meta-analysis of the risk of GERD associated with short sleep duration (*Jansson et al., 2009*; *Lindam et al., 2012*, *2016*; *Zhang et al., 2012b*; *Fass et al., 2005*; *Ju et al., 2013*; *Zhang et al., 2012a*; *Chang et al., 2021*; *Emilsson et al., 2022*; *Ahmed et al., 2020*; *Yadegarfar et al., 2018*).

demonstrated the stability of these results since none of the individual studies reversed the pooled effect size (Fig. S4).

## GERD and risk of Sleep disturbance

Eight articles (*Cadiot et al., 2011*; *Cremonini et al., 2009*; *Ha et al., 2023*; *Hyun, Baek & Lee, 2019*; *Lei et al., 2019*; *Okuyama et al., 2017*; *Wallander et al., 2007*; *You et al., 2015*) were reviewed to determine the risk of sleep disturbance associated with GERD. In spite of a significant OR (OR = 1.99, 95% CI [1.13–3.49], $p < 0.001$), there was a high degree of heterogeneity between the articles ($I^2$ = 99.2%, z = 2.39; Fig. S5), and sensitivity analyses showed that none of the individual studies had a significant impact on the results of the meta-analysis (Fig. S6). It is noteworthy that three (*Cadiot et al., 2011*; *Lei et al., 2019*; *Okuyama et al., 2017*) of these eight articles did not adjust for confounders in the study population during the course of the study. Regression analysis, applied to derive a $p$-value for the comparison between the two groups, was significant ($p = 0.038$). Consequently, only studies adjusting for confounders were analyzed (*Cremonini et al., 2009*; *Ha et al., 2023*; *Hyun, Baek & Lee, 2019*; *Wallander et al., 2007*; *You et al., 2015*). Accounting for adjusting confounders significantly reduced the aforementioned heterogeneity ($I^2$ from 99.2% to 71.6%), and a significant link between GERD and the risk of developing sleep disturbance was re-affirmed (OR = 1.47, 95% CI [1.24–1.74], $p < 0.001$; Fig. 3.2).

## GERD and risk of short sleep duration

An analysis of three articles (*Chen et al., 2009*; *Ha et al., 2023*; *Murase et al., 2014*) examining the correlation between GERD and short sleep duration demonstrated a clear association. There was no significant heterogeneity between the three included articles ($I^2$ = 0, z = 7.79). Therefore, we decided to use fixed effects for our meta-analyses. The pooling analysis shows that a history of GERD corresponds with an increased risk of short sleep duration (OR = 1.17, 95% CI [1.12–1.21], $p < 0.001$; Fig. 3.3). Subsequent sensitivity analyses bolstered these findings, with no individual study significantly influencing the meta-analysis results (Fig. S7).

## Publication bias

Our investigation into publication bias involved examining funnel plots across different subgroups and conducting Egger's regression test for statistical verification. Figure 4 is a funnel plot of the meta-analysis of insomnia and the risk of GERD (*Fass et al., 2005*; *Jansson et al., 2009*; *Ju et al., 2013*; *Lindam et al., 2012, 2016*; *Zhang et al., 2012b*), and the Egger's regression test ($p = 0.038$) also showed no significant publication bias in our meta-analysis. Similar methodology was applied for testing publication bias for additional outcomes, revealing no evidence of bias (Table S2).

## DISCUSSION

This meta-analysis incorporating 22 studies thoroughly addressed the bidirectional relationship between GERD and sleep problems. We found a significant higher risk of poor sleep quality, short sleep duration, or sleep disturbance in individuals with GERD, with the respective risks increased by 1.47-fold, 1.17-fold and 1.47-fold compared to healthy

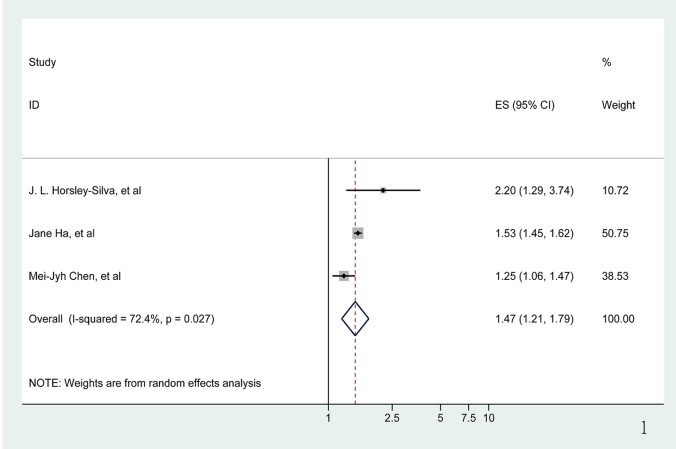

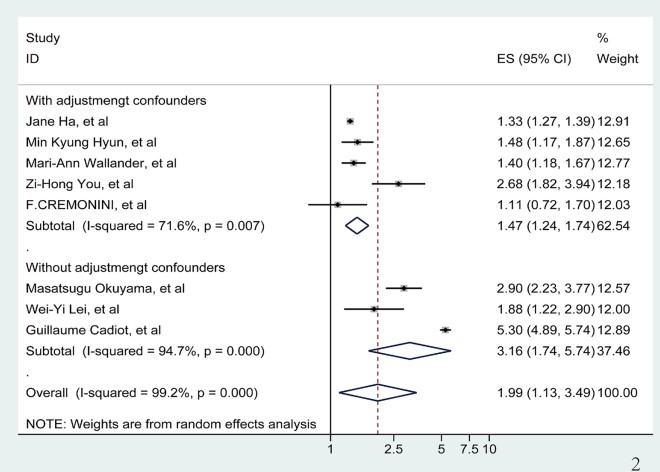

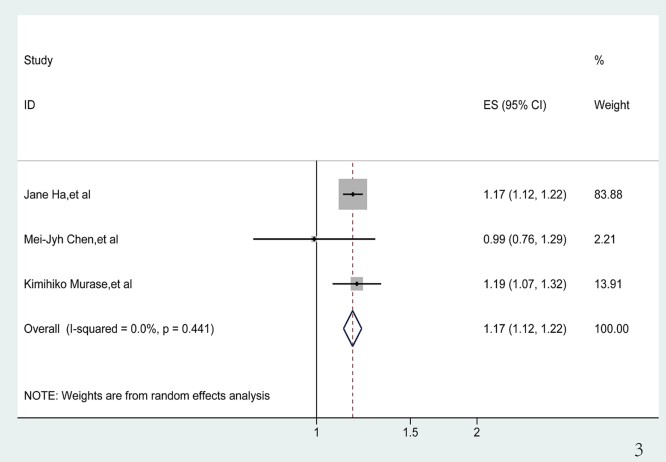

**Figure 3  Meta-analysis of the risk of sleep problems associated with GERD.** (1) Meta-analysis of the risk of poor sleep quality associated with GERD. (2) Meta-analysis of the risk of sleep disturbance associated with GERD. (3) Meta-analysis of the risk of short sleep duration associated with GERD (*Horsley-Silva et al., 2019*; *Ha et al., 2023*; *Chen et al., 2009*; *Hyun, Baek & Lee, 2019*; *Wallander et al., 2007*; *You et al., 2015*; *Cremonini et al., 2009*; *Okuyama et al., 2017*; *Lei et al., 2019*; *Cadiot et al., 2011*; *Ha et al., 2023*; *Chen et al., 2009*; *Murase et al., 2014*).

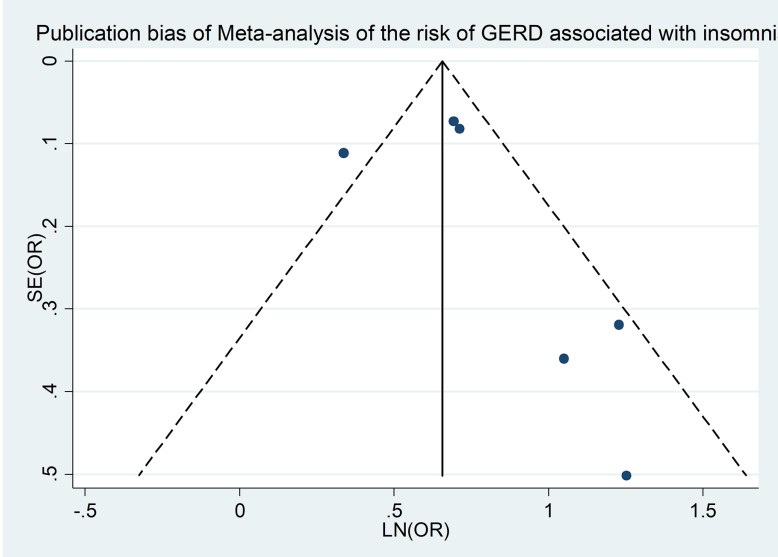

**Figure 4 Publication bias of meta-analysis of the risk of GERD associated with insomnia.**

counterparts. Meanwhile, the risk of GERD is notably higher in those with insomnia, short sleep duration, or sleep disturbances, with risks higher by 1.5, 2.66, or 1.98, respectively. These findings highlight the significance of early recognition of GERD and its sleep-related comorbidities for better clinical outcomes.

A previous meta-analysis examined the relationship between sleep problems and their comorbidities, including GERD (*Huang et al., 2022*). The study with a focus on first responders, showed insomnia increased the risk of depression and anxiety. Though it included GERD data, it lacked definitive insights on the link between insomnia and GERD, potentially due to the targeted population. To fill this research void, we expanded our scope and performed a more comprehensive analysis based on types of sleep problems, substantiating a strong association between insomnia, short sleep duration, sleep disturbance, and GERD. Another prior review posited GERD as a potential indicator of insomnia and sleep initiation issues (*Jung, Choung & Talley, 2010*). However, it did not conclusively infer an increased risk for these sleep problems. In contrast, our statistical analysis clearly revealed that GERD indeed increases the risk of poor sleep quality, short sleep duration, and sleep disturbance.

Both GERD and sleep problems are common. A cross-sectional study of 11,685 GERD patients found them more susceptible to sleep problems (*Mody et al., 2009*). Another previous case-control study showed that patients with sleep problems had significantly higher rates of reflux symptoms than healthy people (*Orr et al., 2008*). GERD is a multifactorial chronic condition with symptoms arising from gastric content reflux. The 24-h esophageal pH test, used in GERD diagnosis (*Gyawali et al., 2018*), shows patients typically exhibiting increased acid reflux frequency, reduced esophageal impedance, and prolonged mucosal recovery time (*Woodland et al., 2013*). Its pathogenesis involves esophagogastric junction incompetence, acid erosion, helicobacter pylori

infection, hiatal hernia, and chronic inflammation (*Katzka & Kahrilas, 2020*). These factors together suggest that a single conventional theory cannot fully account for the coexistence of GERD and sleep issues. Therefore, we propose instead that a multitude of pathogenic mechanisms contribute to their concurrent emergence.

The underlying mechanisms of the reciprocal influence between gastroesophageal reflux disease and sleep disturbances remains poorly understood. The TRPV1 and melatonin pathways may play significant roles in this interplay. TRPV1, an acid-sensitive receptor, is activated by both capsaicin and heat and is present in the esophageal mucosa epithelial cells, which produces a burning sensation during acid reflux (*Ma et al., 2012*). Studies have identified TRPV1 expression in the hypothalamus (*Jeong et al., 2018*). Research by *Liu & Tian (2023)* elucidated TRPV1's involvement in sleep-wake cycles through experiments involving capsaicin administration in animal subjects. Sustained acid-mucosal contact may have an initiating effect on central nervous system arousal mechanisms. At the same time, evidence suggests that poor sleep quality can exacerbate reflux incidents and increase acid contact time (*Hung et al., 2016*). Total sleep deprivation has been shown to induce esophageal hyperalgesia, a condition observable in the acid perfusion test (*Onen et al., 2001*). This acid reflux abnormality can induce esophageal pain and consequently disrupt sleep, while simultaneous sleep deprivation can intensify esophageal sensitivity and aggravate this effect. Hormonal changes could influence both GERD and sleep issues. It is well known that sleep problems can directly affect sleep rhythms. Melatonin, derived from L-tryptophan, is synthesized in the pineal gland and operates under the regulation of sleep rhythms (*Majka et al., 2018*). This hormone acts to reduce transient lower esophageal sphincter relaxations by suppressing nitric oxide biosynthesis, thus potentially mitigating GERD morbidity (*Pereira, 2006*), which may elucidate the link between sleep problems and GERD. Furthermore, psychological aspects are influential; GERD symptoms could predispose individuals to psychiatric conditions, including depression (*Núñez-Rodríguez & Miranda Sivelo, 2008*). In cases of GERD, mucosal damage results from a combination of inflammatory and immune factors (*Kandulski & Malfertheiner, 2011*), both of which have been implicated in depression (*Slavich & Irwin, 2014*). A meta-analysis puts forward that sleep problems can double the risk of depression or even herald its onset (*Baglioni et al., 2011*). Thus, depression might act as a mediator between GERD and sleep issues, with inflammation being a significant contributor. In summary, the association between GERD and sleep disturbances is complex and mutual, challenging simple explanations offered by traditional theories.

## IMPLICATIONS AND LIMITATIONS

Our study synthesizes the existing evidence on the relationship between GERD and sleep problems, demonstrating their bilateral influence. It emphasizes the need to consider the risk of sleep problems in patients with GERD as well as recognizing that those with sleep problems are more prone to GERD symptoms. These conclusions inform clinical practice. Confirming the bidirectional association between GERD and sleep problems offers a foundational basis for further clinical research. Future studies could explore the causative factors underlying the relationship between GERD and sleep problems. Healthcare

providers should be aware that GERD may coexist with sleep problems, prompting consideration for combined treatment strategies to improve therapeutic outcomes.

Nonetheless, this study is not without limitations. The use of multiple diagnostic criteria for sleep problems introduces variability, and future studies should strive for standardized inclusion criteria. The study's use of clinical symptoms as inclusion criteria, rather than objective clinical examination, may have resulted in the exclusion of some patients who were asymptomatic. The inclusion of cohort and case-control studies instead of randomized controlled studies may lead to heterogeneity. Perhaps due to the presence of well-defined diagnostic criteria and clinical indicators for obstructive sleep apnea (OSA), as opposed to other sleep problems diagnosed primarily through medical history and questionnaires, there is a scarcity of research concurrently investigating gastroesophageal reflux, OSA, and other sleep problems. Therefore, this study did not include OSA as a focal point of investigation. In addition, we did not include covariate analysis in this study. Although most of the literatures we included had been adjusted for confounders, differences in the adjustment for confounders between the articles are still likely to have an impact on the results. While previous studies have established an association between GERD and esophageal hiatal hernia (*Jones et al., 2001*), we did not incorporate it as a confounding factor due to insufficient relevant data, potentially introducing bias. Finally, some articles in the meta-analysis scored lower on quality assessment. Future research should include high-quality prospective cohort studies to ensure more reliable results.

## CONCLUSION

This meta-analysis indicates that GERD elevates the risk of insomnia, short sleep duration, or sleep disturbance. Conversely, poor sleep quality, short sleep duration, or sleep disturbance independently pose a risk for GERD. Our findings underscore the importance for healthcare practitioners to be vigilant regarding the correlation between GERD and sleep disturbances in clinical settings.

### Funding
The authors received no funding for this work.

### Competing Interests
The authors declare that they have no competing interests.

### Author Contributions
- Xiaolong Tan conceived and designed the experiments, performed the experiments, analyzed the data, prepared figures and/or tables, authored or reviewed drafts of the article, and approved the final draft.
- Shasha Wang performed the experiments, prepared figures and/or tables, and approved the final draft.
- Fengjie Wu analyzed the data, prepared figures and/or tables, and approved the final draft.

- Jun Zhu conceived and designed the experiments, authored or reviewed drafts of the article, and approved the final draft.

## Data Availability

This is a systematic review/meta-analysis.

## Supplemental Information

Supplemental information for this article can be found online at http://dx.doi.org/10.7717/peerj.17202#supplemental-information.

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
