# Peer review of "Bidirectional correlation between gastroesophageal reflux disease and sleep problems: a systematic review and meta-analysis"

_PeerJ, doi:10.7717/peerj.17202_

## Round 0.1 · original submission · Minor Revisions

I have now received the reviewers' comments on your manuscript. They have suggested some revisions to your manuscript. Therefore, I invite you to respond to the reviewers' comments and revise your manuscript.

Reviewer 1 ·

Basic reporting

In the present systematic review (SR) and meta-analysis (MA) the Authors hypothesized a potential two-way connection between gastroesophageal reflux disease (GERD) and sleep problems. Thus, they systematically reviewed population-based evidence to ascertain their relationship. The Authors found that GERD increases the risk of insomnia, short sleep duration, or sleep disturbance. Besides, poor sleep quality, short sleep duration, or sleep disturbance are also independent risk factor for GERD.
Overall, I found the present SR/MA timely, very interesting, original, well conducted and scientifically sound. However, I have some comments aimed at improving the quality of the paper, and these are outlined below:
• In the introduction, a brief note on the fact that GERD and sleep problems might be linked and comorbid with several psychiatric disorders (with the probable involvement of melatonergic system indeed and the possible therapeutic effect of the drug agomelatine), should be added with appropriate references (please see and refer to following dois: 10.1016/j.amsu.2022.104056, 10.3390/ijms140612458 and 10.2174/187152711794488674).
• Translating into “real world” clinical practice and medicine, what possible clinical shreds of evidence might arise from the present study and what the Researchers do suggest to improve practice? Please add a brief paragraph on possible suggestions in terms of integrative care.
• Moreover, what the Authors suggest and recommend to clinicians, stakeholders and policy makers to face this complex problem? In particular, in conclusion the Authors wrote that “The results of our meta-analysis facilitate the development of new ideas for the treatment of GERD and sleep problems”. Nevertheless, what are these ideas?
• Referring to the previous point, translating into “real world” clinical practice and medicine, what possible clinical shreds of evidence might arise from the present SR/MA and what the Researchers do suggest to improve practice? Please add a brief résumé paragraph on “recommendations” in terms of integrative care.
• Finally, I suggest improving the English language with the help of a native speaker as there are some grammatical errors and typos.

Experimental design

Please, see above

Validity of the findings

Please, see above

Additional comments

Please, see above

·

Basic reporting

Tan et al. present the results of a well prepared pairwise meta-analysis evaluating the bidirectional association between GERD and four sleep disorders (i.e., sleep disturbance, short sleep duration, insomnia, and poor sleep quality). A total of 22 studies published up to Aug/2023 were included in the review. Results are expressed as ORs, and the authors conclude that GERD increases the risk of poor sleep quality, sleep disturbance, and short sleep duration, as well as insomnia, sleep disturbance, and short sleep increase the risk of presenting GERD. Overall, this is an interesting analysis and approach to the research question - the author should be congratulated for their efforts; however, the manuscript would benefit from some minor clarifications/editions.

Experimental design

-The PROSPERO register "CRD42023452348" is congruent with the design of the SLR and metanalysis presented in this manuscript; however, the last update in the NIHR database was done in August/2023 - the authors should update the information in PROSPERO.
-Why including only case-control or cohort studies? any specific reason? why not including RCTs? there was any pre-screening proccess before study selection?
-A graphical representation (using an appropriate color scale) may benefit the supplementary material presenting the risk assessment rather than only the NOS; moreover, if possible, it would be adequate to include the level of evidence for each study using a valid tool (i.e., the OCEBM Levels of Evidence).
-During the study selection process, please clarify how was the decission of inclusion taken (agreement between both reviewers?, decision by one of the two reviewers? ... in case of disagreement was there a third reviewer? was a blinded process? etc...)
-For the data extraction, how was the data tabulated in an Excel spreadsheet? Other data capture tools? - which data points for each study were extracted? - I believe that this information is on the protocol registered at PROSPERO, but this should be included in the manuscript as well.
- Statistical models to perform the meta-analysis are overall adequate; the presentation of results from random effects models is consistent with the proposed statistical analysis.
-Please clarify in line 55 what the definition of GERD as inclusion criteria for this review (saying that diagnosis is made by a questionnaire or endoscopy is unclear); moreover, it would be beneficial if the Table presented the summary of each of the included studies the authors include two columns describing what was the definition of GERD (i.e. diagnostic method) and the diagnostic methods for the indicated sleep disorder.
-It would be beneficial to include/specify the PICO question in the Methods so the readers can understand the purpose of the analysis faster.
-Why limit this analysis only to 4 sleep disorders?; in the abstract authors highlight the potential relationship with other related disorders such as OSA - would suggest is possible including this disorder into analysis as well.

Validity of the findings

-In the introduction, it will be desirable that the authors write a paragraph summarizing the scientific rationale of the two-way connection between GERD and sleep problems and the current hypothesis; also, an important question that should be addressed is: Is the presence of Hiatal Hernia a key factor (i.e., contributor factor) in patients with GERD to present sleep disorders? - I suggest considering the role of HH in the discussion as well as the limitations of the study. That factor was not adjusted/considered as cofounder in most of the included studies - any comment on that?
-I strongly encourage the authors to provide open access to the data used in this Analysis. A link to an institutional repository in which the file with the collected data can be consulted at any point is highly desirable.

Additional comments

-In the abstract, I suggest starting with a short background mentioning the potential relationship between GERD and sleep disorders rather than with the objective of the meta-analysis.
-In the discussion separate paragraphs (lines 183 and 184), I making a better transition between topics before explaining the potential theories and mechanisms, including TRPV1 and melatonin pathways.
I suggest unifying (making homogeneous) the report on Age in Table 1 and Table 2. Maybe the use of some indicators or footnotes will make the data clearer and easier to read.
-For figures 2 and 3, I suggest using rounded numbers with one decimal in the X-axis of the forest plots and the use of regular intervals instead of automatic adjustment (e.g., Figure 2.1 instead of 0.107, 1, 9.36 using 0, 2.5, 5, 7.5, 10 will be better), this also will allow the beginners in interpreting meta-analysis to have a more realistic picture of the results (without graphical representation adjustments).
-Please check all the reference numbers and order of presentation; there are some incongruences (e.g., see, reference list lines 230, 254, 258, 261, 272, etc..]
Figure 1, which presents the screening process, has a poor resolution/quality. It is recommended that the file be reviewed.
In Figure 4, it would be beneficial to indicate which study corresponds to each point of the funnel plot.
Some minor grammatical/redaction errors were identified, but I guess this can be addressed in the editorial process.

Reviewer 3 ·

Basic reporting

-

Experimental design

-

Validity of the findings

-

Additional comments

I would like to thank you for inviting me to review this manuscript. I read the article carefully. This meta-analysis aims to explore the association between sleep problems and gastroesophageal reflux disease (GERD). I believe that the methods, results and discussion sections are well written and well referenced. However, in the introduction section, it seems necessary to mention more details about the pathophysiology of the bidirectional relationship between sleep disorders and GERD. More explanations in this regard can prepare the reader's mind for a deeper understanding of the meta-analysis results. All in all, it is a good work.

---

## Round 0.2 · Minor Revisions

Thank you for the update. However, there are still concerns that prevent me from accepting the revised paper. Please pay attention to the reviewers' comments and respond to them carefully.

·

Basic reporting

I appreciate the time and effort of the authors to carefully review each comment. I believe that the manuscript is now in better shape, and this publication will open interesting avenues for discussion regarding the unexplored relationship between sleep disorders and GERD. A minor observation is that the definition used in this review was adopted from the Montreal consensus, which is based on clinical/endoscopic features, "troublesome symptoms and/or complications." this may cause some limitations in the interpretations due to the exclusion of several patients (or studies) reporting with lack of "troublesome symptoms" but objective evidence of abnormal acid exposure (i.e., pH monitoring). Current definitions and diagnostic criteria of GERD require some evidence of conclusive reflux-related pathology on endoscopy and/or abnormal pH monitoring (see Lyon Consensus v2.0) - this should be adressed as a study limitation. Finally, I would suggest the use of professional language services due to the significant number of errors noted in the reviewed version. Once again, I thank the authors for their corrections and congratulate them for their efforts.

Experimental design

Comments were made in the original review, the authors adressed most of them. Please refer to the letter of response.

Validity of the findings

Comments were made in the original review, the authors adressed most of them. Please refer to the letter of response.

Additional comments

N/A

---

## Round 0.3 · accepted · Accept

In my opinion this manuscript has been revised with attention to the reviewers' comments and can now be published.

·

Basic reporting

I appreciate the changes made by the authors. I believe the manuscript is suitable for publication in the Journal. I would suggest a last language proof by the editor or the publisher. Thanks for let me review this manuscript.

Experimental design

N/A

Validity of the findings

N/A

Additional comments

N/A